# Do Lifestyle Interventions before Gastric Bypass Prevent Weight Regain after Surgery? A Five-Year Longitudinal Study

**DOI:** 10.3390/nu14173609

**Published:** 2022-08-31

**Authors:** Salvatore Vaccaro, Leila Itani, Francesca Scazzina, Stefano Bonilauri, Concetto Maria Cartelli, Marwan El Ghoch, Massimo Pellegrini

**Affiliations:** 1Clinical Nutrition Unit, Azienda Unità Sanitaria Locale—IRCCS di Reggio Emilia, 42123 Reggio Emilia, Italy; 2Human Nutrition Unit, Department of Food and Drug, University of Parma, 43125 Parma, Italy; 3Department of Nutrition and Dietetics, Faculty of Health Sciences, Beirut Arab University, P.O. Box 11-5020 Riad El Solh, Beirut 11072809, Lebanon; 4General and Urgent/Emergency and Bariatric Surgery Unit, Azienda Unità Sanitaria Locale—IRCCS di Reggio Emilia, 42123 Reggio Emilia, Italy; 5Department of Biomedical, Metabolic and Neural Sciences, University of Modena and Reggio Emilia, 41125 Modena, Italy

**Keywords:** bariatric surgery, lifestyle modification, obesity treatment, weight loss maintenance

## Abstract

It is unclear whether weight loss (WL) achieved by means of lifestyle interventions (LSIs) before bariatric surgery (BS) can improve long-term WL outcomes after surgery. We aimed to assess the impact of a structured LSI on WL% after gastric bypass (GBP). Two groups of patients were selected from a large cohort of participants with obesity who underwent GBP surgery at Santa Maria Nuova Hospital (Reggio Emilia, Italy). The groups were categorized as those who have or have not received LSI prior to GBP. The LSI group included 91 participants (cases) compared to 123 participants (controls) in the non-LSI group. WL% was measured at follow-up times of 1, 3, 6, 12, 24, 36, 48, and 60 months. The LSI group achieved a clinically significant WL% (−7.5%) before BS, and at the time of surgery, the two groups had similar body weights and demographic statuses. At all points, until the 24-month follow-up, the two groups displayed similar WLs%. With regard to the longer follow-ups, the LSI group maintained weight loss until the last timepoint (60 months), whereas the non-LSI group experienced weight regain at 36, 48, and 60 months. In a real-world context, a structured behavioral LSI prior to GBP seems to prevent longer-term weight regain.

## 1. Introduction

Obesity is a chronic disease characterized by an increased deposition of fat mass, known to be associated with a higher risk of medical and psychological comorbidities, physical and social disabilities, and mortality [1]. In the last few decades, an increase in prevalence has been observed globally across different age groups and genders, which is considered an epidemic [2]. This requires a wide range of weight loss interventions, including lifestyle interventions (LSIs), pharmacotherapy, and bariatric surgery (BS) [3]. Despite these interventions, regardless of the type of intervention, weight regain is common in patients with obesity who have intentionally lost weight [4], yet the drivers of relapse are not fully understood [5]. Therefore, two key factors in ensuring the success of weight management treatments in relation to obesity are the identification of factors that lead to the lack of long-term weight loss maintenance and the implementation of effective strategies to prevent weight regain [6].

While LSIs are considered the key elements and the bases of weight management for patients with obesity [7], BS still remains the most effective intervention [8]. However, patients who undergo this type of treatment are not immune to weight regain, especially in the long term [9]. In this context, systematic reviews and meta-analysis data have demonstrated that receiving LSIs—based on behavioral therapy for obesity (BT-OB)—after surgery determined greater WL than usual care [10]. However, the hypothesis that the same LSI delivered before BS can improve WL outcomes after surgery has not been confirmed [11,12]. A recent systematic review showed no conclusive evidence that intentional WL before BS improved post-operative WL outcomes because of methodological heterogeneity (i.e., not always LSI) and variation in the duration of post-surgical follow-up [13].

The current study, therefore, aimed to investigate the relationship between pre-operative WL achieved by means of a structured LSI and short-term (1-month, 3-month, and 6-month), intermediate (12-month and 24-month), and long-term (36-month, 48-month, and 60-month) WL outcomes after BS in a real-world clinical setting.

## 2. Materials and Methods

### 2.1. Participants and Study Design

This is a single-center retrospective observational study. Participants were selected from a large cohort previously enrolled in the BS pathway at the Santa Maria Nuova Hospital of Reggio Emilia (Italy) between January 2006 and December 2012. A total of 91 patients assessed for eligibility were included because they fulfilled all of the following conditions: (i) were aged between 20 and 70 years; (ii) were enrolled on, attended, and completed an LSI program prior to BS; (iii) underwent a gastric bypass (GBP) technique as BS; (iv) successfully completed all follow-up assessments. The LSI was proposed to all patients on the waiting list after being revealed to be appropriate for BS for logistic issues (i.e., patients living near Reggio Emilia and those with occupational flexibility were more inclined to attend the LSI program). Patients unable to attend the LSI program were left on the waiting list until the date of BS.

The LSI was composed of 12 sessions held over six months (one individual and one group session per month), led by a clinical dietitian and psychologist. It was based on BT-OB with the adoption of a healthy lifestyle and obesity-reducing behaviors, favoring the development of a positive energy balance through a personalized low-calorie Mediterranean diet and regular physical activity [14]. The topics of the sessions that were focused on include: (1) monitoring food intake; (2) changing eating habits; (3) developing an active lifestyle; (4) weight loss goals; and (5) body image. The control group contained 123 patients from the same cohort with a similar BMI and the same gender distribution as the intervention group, who were assigned to the waiting list (non-LSI) with a 1:1 ratio. The study was approved by the Local Ethics Committee of Reggio Emilia (STATNUTRIZ14_1 no. 10/2105) on 21 January 2015. All personal data of the patients were treated according to European/Italian privacy laws.

### 2.2. Times of Assessment of Follow-Up

T-PreLSI = the time before BS, T-0 = the time at BS, T-1 = 1-month follow-up, T-3 = 3-month follow-up, T-6 = 6-month follow-up, T-12 = 12-month follow-up, T-24 = 24-month follow-up, T-36 = 36-month follow-up, T-48 = 48-month follow-up, and T-60 = 60-month follow-up after BS.

### 2.3. Baseline Measures

The socio-demographic variables (age, sex, marital status, and employment) were retrieved from baseline anamnestic records.

Body weight and height at baseline (T-0) were measured using an electronic weighing scale (Wunder MPMP-DP2400, Wunder, Trezzo sull’Adda (MI), Italy) and a wall-mounted stadiometer (Wunder, Trezzo sull’Adda (MI), Italy). BMI was calculated according to the standard formula of body weight in kilograms divided by the square of the height in meters.

Cardiometabolic disease in this study was retrieved from baseline anamnestic records and defined as the presence of any diseases, such as type 2 diabetes, hypertension, and dyslipidemia (lowered level of high-density lipoprotein cholesterol and an increased level of low-density lipoprotein cholesterol and triglycerides).

### 2.4. Bariatric Surgery Procedure

GBP is currently the most widely used procedure for surgical treatment of obesity. It is a restrictive-malabsorptive procedure [15] that traditionally consists of making a small stomach pouch (15–30 cc) directly connected to the small intestine at a variable distance from the duodenum [16]. The main remaining part of the stomach and the duodenum were completely excluded from the transit of food. The procedure did not involve the removal of any part of the intestine or stomach and is considered reversible [16]. The procedure takes about two to three hours and requires a hospital stay of four to eight days.

### 2.5. Follow-Up Measures

Body weight was measured, and WL% was determined at all times of follow-up: WL% = [(body weight at T-0 − body weight at T-1, 3, 6, 12, 24, 36, 48, 60)/body weight at T-0] × 100.

### 2.6. Statistical Analysis

Descriptive statistics are presented as means and standard deviations for continuous variables, and as frequencies and proportions for categorical variables. Student’s t-test was used to compare means, and a chi-squared test of independence was used for proportions. A mixed two-way analysis of variance (ANOVA) was used to conduct repeated measure analysis, with time and intervention group before BS as the independent variables and WL% as the dependent variable. The main effects of time, intervention, and their interaction were determined with mixed ANOVA. When the sphericity assumption was not met for mixed ANOVA, the Geisser–Greenhouse correction F-test was used [17]. The significance level used for all tests was *p* < 0.01. A posteriori power analysis for repeated measures ANOVA was conducted using G*Power version 3.1.2.9, which revealed a 100% power for a sample size of 214 for two groups at an alpha level of 0.01 [18].

## 3. Results

The study sample consisted of 214 participants, mostly female (79%), with a mean age of 43.95 ± 10.83 years. The mean age (43.59 ± 10.36 vs. 44.22 ± 11.21 years) and sex distribution (75% vs. 82.2% female) were similar between the two groups. Similarly, participants in both groups did not differ with respect to marital (85.7% vs. 89.4% married) or employment status (92.3% vs. 93.5% employed). As for cardiometabolic disease, the prevalence of type 2 diabetes (15.4% vs. 13.8%), hypertension (37.4% vs. 28.5%), and dyslipidemia (7.7% vs. 3.3%) was similar in both groups (Table 1). Participants who joined the LSI group were heavier (132.90 ± 22.96 vs. 122.10 ± 19.22 Kg) at the T-Pre-LSI and experienced a clinically significant WL% (−7.50 ± 3.79%) before BS compared to those who did not follow the LSI (−1.23 ± 5.08%). Both groups had a similar weight (122.98 ± 22.04 vs. 120.72 ± 20.30 Kg) and BMI (45.16 ± 6.36 vs. 45.02 ± 5.90) before BS at T0.

The mixed ANOVA showed that there was a significant main effect of intervention (F(1, 212) = 12.3, *p* = 0.001) and time (F(1.84, 389.6) = 986.3, *p* < 0.001) on WL%, and a significant interaction between time and intervention type before BS (F(1.84, 389.6) = 88.0, *p* < 0.001). The pairwise analysis (Table 2 and Figure 1) comparing the mean WL% at each point in time between the two intervention groups showed that the mean WL% between both groups did not differ significantly at 1 month (−8.41 ± 3.06% vs. −9.70 ± 4.74%), 3 months (−15.81 ± 4.80% vs. −17.13 ± 5.76%), 6 months (−22.89 ± 6.02% vs. −23.76 ± 7.20%), 12 months (−29.66 ± 7.75% vs. −29.31 ± 8.79%), or 24 months (−33.13 ± 7.71% vs. −30.43 ± 8.88%) post-BS, despite the significant intervention type main effect.

The significant interaction between time and intervention type BS was reflected in a significantly different WL% between groups at 36 months (−34.18 ± 7.91% vs. −28.61 ± 9.22%), 48 months (−36.25 ± 7.50% vs. −27.32 ± 9.62%), and 60 months (−35.42 ± 7.37 vs. −23.55 ± 9.87%) post-BS (Table 2, Figure 1). Therefore, with the progression of time post-BS, specifically after 36 months, individuals receiving an LSI were able to maintain their WL compared to those in the non-LSI group. Pairwise comparisons showed that participants in the non-LSI group failed to maintain WL% and reverted at 60 months (−23.55 ± 9.87%) to the WL% they achieved at 6 months (−23.76 ± 7.20%) post-BS (*p* = 1.00). Those in the LSI group maintained a significantly increasing WL% throughout the follow-up period from 6 months (−22.89 ± 6.02%) to 12 months (−29.66 ± 7.75%) and 24 months (33.13 ± 7.71%) and maintained the same WL% after 36 months (−34.18 ± 7.91), 48 months (−36.25 ± 7.50%), and 60 months (−35.42 ± 7.37%). The WL% was significantly different (*p* < 0.001) across all time points in the LSI group post-BS.

## 4. Discussion

The present study aimed to investigate the effect of intentional WL from a structured LSI prior to BS on the short-term, intermediate, and long-term WL outcomes after BS. The main finding showed that patients who underwent the LSI prior to BS achieved a clinically significant WL (7.5%) [19]. However, compared with those in the non-LSI group, they did not experience higher WL in the short-term (1-month, 3-month, and 6-month) or intermediate (12-month and 24-month) follow-up. After a longer period of follow-up (36, 48, and 60 months), the WL was maintained in the LSI group, while the non-LSI group experienced significant weight regain from month 24 to month 60.

The initial reading of this finding may be that it appears to contradict the available literature, which found that intentional WL prior to BS seems not to be effective in improving WL outcomes after BS [13,20,21,22]. There are two important issues that may explain this discrepancy. First, the intentional WL reported in some of the previous studies was not always achieved by means of a structured LSI but by prescriptive diets (i.e., two weeks of a very low calorie diet (VLCD)) [23,24] or intra-gastric balloon (IGB) [25]. Therefore, the strategies of LSIs associated with the successful adoption of obesity-reducing behaviors that give the patient skills to enable them to manage their body weight in the long term are lacking [14]. Second, the studies that delivered structured LSIs based on BT-OB [26,27] or CBT-OB [28] and found no superiority of LSI in terms of WL post-BS only had a maximum 24-month follow-up period. In our study, we did not detect any difference in WL before this point, and the impact of the LSI was seen from 36 months.

The reason why this effect was only seen in the later stages is unclear, but we can speculate that the strong restrictive and malabsorptive effect of GBP seems to mask any other factor (i.e., the treatment LSI program). BS patients do not usually face substantial obstacles in losing weight in the short-to-intermediate term, but difficulties with weight regain are noticed in later stages (two to three years later), especially in GBP [9,29]. At this later point, the effect of BS seems to lessen, with the re-adoption of unhealthy behaviors (i.e., increased food intake and reduced physical activity). Patients who received an LSI developed skills to use as strategies to manage their body weight better than those who were not enrolled in an LSI.

Our study has several strengths. To the best of our knowledge, it is one of only a few studies to assess the relationship between pre-operatory WL in BS treatment-seeking patients with obesity who underwent a well-structured LSI in comparison to a group who did not. The longitudinal controlled design over a long follow-up period in a real-world clinical setting should also be considered strengths. There are also a number of limitations. Firstly, data were obtained from a single clinical unit that applied GBP, and external validation is required in other populations and BS techniques (i.e., sleeve gastrectomy, etc.) [30]. Secondly, patients whose follow-up assessment was interrupted were disregarded, meaning that intent-to-treat analysis was not conducted, which may limit generalizability [31]. Thirdly, the lack of information on an objective assessment of body circumferences (i.e., waist, hip, etc.) and body composition, such as fat mass or fat-free mass measured with bioelectrical impedance analysis (BIA) and dual-energy X-ray absorptiometry (DXA) at baseline and follow-up times. Finally, the relatively small sample size and the retrospective design [32]. This means that these results are preliminary, and in order to avoid any bias, they need further replication through prospective studies (i.e., randomized controlled trials) with larger samples including a wide range of variables (metabolic, body composition, biochemical, etc.).

Our findings have a number of implications. First, policymakers should take our results at least as preliminary evidence to implement LSIs before surgery in the routine BS pathway. Second, awareness should be raised among all healthcare professionals dealing with obesity in surgical settings to motivate and engage their patients seeking BS treatment to adhere to these LSIs, while explaining the positive clinical impact after surgery in terms of WL.

## 5. Conclusions

Our study provides evidence for the beneficial effect of a structured LSI based on BT-OB prior to BS. Therefore, it is worth establishing new guidelines that recommend the inclusion of LSIs in the preparation phase prior to BS since patients who adhere to such programs seem to have better WL outcomes in the longer term (five years later). This type of program should be considered a useful strategy to improve BS outcomes.

## Figures and Tables

**Figure 1 nutrients-14-03609-f001:**
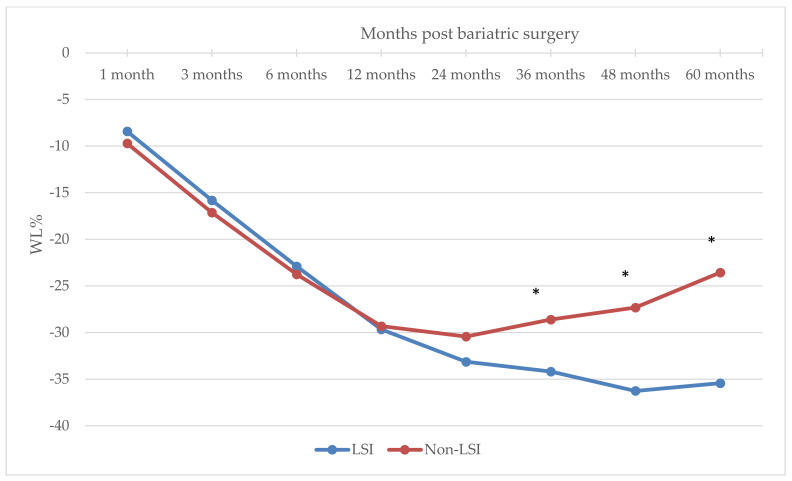
Change in WL% over time in the LSI and Non-LSI groups. * Significant at *p* < 0.01.

**Table 1 nutrients-14-03609-t001:** Characteristics of the study participants before BS.

	LSI *n* = 91	Non-LSI *n* = 123	Total *n* = 214	Significance
Age (years)	43.59 ± 10.36	44.22 ± 11.21	43.95 ± 10.83	*p* = 0.670
Sex				X^2^ = 1.72, *p* = 0.190
Male	23(25.3)	22(17.9)	45(21.0)	
Female	68(74.7)	101(82.1)	169(79.0)	
Marital Status				X^2^ = 0.677, *p* = 0.411
No	13(14.3)	13(10.6)	26(12.1)	
Yes	78(85.7)	110(89.4)	188(87.9)	
Employment				X^2^ = 0.113, *p* = 0.736
No	7(7.7)	8(6.5)	15(7.0)	
Yes	84(92.3)	115(93.5)	199(93.0)	
Height (cm)	1.65 ± 0.08	1.64 ± 0.09	1.64 ± 0.09	*p* = 0.317
Weight (Kg)	122.98 ± 22.04	120.72 ± 20.30	121.70 ± 21.04	*p* = 0.444
BMI (Kg/m^2^)	45.16 ± 6.36	45.02 ± 5.90	45.08 ± 6.08	*p* = 0.862
Hypertension				X^2^ = 1.90, *p* = 0.168
No	57(62.6)	88(71.5)	145(67.8)	
Yes	34(37.4)	35(28.5)	69(32.2)	
Type 2 Diabetes				X^2^ = 0.103, *p* = 0.748
No	177(84.6)	106(86.2)	183(85.5)	
Yes	14(15.4)	17(13.8)	31(14.5)	
Dyslipidemia				X^2^ = 2.12, *p* = 0.146
No	84(92.3)	119(96.7)	203(94.9)	
Yes	7(7.7)	4(3.3)	11(5.1)	

LSI = lifestyle intervention; BMI = body mass index.

**Table 2 nutrients-14-03609-t002:** WL% at different time points among participants in the two groups (*n* = 214).

Months Post-Bariatric	LSI *n* = 91	Non-LSI *n* = 123	Significance ^¥^
		% Weight loss	
1 month	−8.41 ± 3.06	−9.70 ± 4.74	0.025
3 months	−15.81 ± 4.80	−17.13 ± 5.76	0.077
6 months	−22.89 ± 6.02	−23.76 ± 7.20 ^a^	0.348
12 months	−29.66 ± 7.75	−29.31 ± 8.79 ^b^	0.758
24 months	−33.13 ± 7.71	−30.43 ± 8.88	0.021
36 months	−34.18 ± 7.91	−28.61 ± 9.22 ^b^	<0.0001
48 months	−36.25 ± 7.50	−27.32 ± 9.62	<0.0001
60 months	−35.42 ± 7.37	−23.55 ± 9.87 ^a^	<0.0001

^¥^ Significant at *p* < 0.01; ^a,b^ values with similar superscripts are significantly different across time at *p* < 0.01; LSI = lifestyle intervention.

## Data Availability

The dataset in the present study is available upon request.

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
