# Peer review of "Do Lifestyle Interventions before Gastric Bypass Prevent Weight Regain after Surgery? A Five-Year Longitudinal Study"

_nutrients, 2022, doi:10.3390/nu14173609_

Round 1
Reviewer 1 Report
The paper addresses the important issue of psychological preparation of patients for bariatric surgery and coping with its consequences after it is performed in order to achieve the planned treatment goals. Certainly, the research should be continued with a larger number of patients. Ultimately, it would be worthwhile to establish guidelines / recommendations taking into account the proper preparation of patients for bariatric surgeries and dealing with their effects on the basis of the conducted research.
Author Response
The paper addresses the important issue of psychological preparation of patients for bariatric surgery and coping with its consequences after it is performed in order to achieve the planned treatment goals. Certainly, the research should be continued with a larger number of patients. Ultimately, it would be worthwhile to establish guidelines/recommendations taking into account the proper preparation of patients for bariatric surgeries and dealing with their effects on the basis of the conducted research.
Response: Done as suggested, and added accordingly in the Discussion (Page 6, paragraph 4) and Conclusions (Page 7, paragraph 1) sections.
Reviewer 2 Report
The authors in this paper aimed to assess the impact of a structured LSI on long-term weight loss after gastric bypass (GBP). As a conclusion they provided evidence for the beneficial effect of a structured LSI prior to BS. The manuscript taken together is well structured and easy to read. However, the authors should kindly provide additional data to support further their results:
Major points:
1) Once talking about obesity and success of therapeutic approaches (surgical or life style interventions), then Waist circumference is an important factor that can not be ignored. Weight or BMI is not sufficient for many reasons described in details in literature. Hence, the authors should include Waist of both groups and baseline and at end of follow-up.
2) The baseline table looks somehow `poor of data´. In other words, the authors are kindly invited to give more additional data in both group at baseline as presence of metabolic syndrome, Diabetes, hypertension, dyslipidaemia, alcohol consume, smoking status, socioeconomic status of both groups. The presence of differences should be then possibly in a multivariate analysis included. Such information is essential to avoid Bias when comparing the both treatment strategies.
3) Since the analysis was focused on weight loss as absolute value then it is expected to have analysis of body composition (measured by BIA or DXA). Therefore what was the value of fat mass or fat free mass in these two categories at the start point and end of follow-up. This point is also essential to avoid false conclusions about weight loss.
Author Response
Reviewer 2
The authors in this paper aimed to assess the impact of a structured LSI on long-term weight loss after gastric bypass (GBP). As a conclusion they provided evidence for the beneficial effect of a structured LSI prior to BS. The manuscript taken together is well structured and easy to read. However, the authors should kindly provide additional data to support further their results:
Response: We thank the reviewer for her/his valuable comments.
Major points:
1) Once talking about obesity and success of therapeutic approaches (surgical or life style interventions), then Waist circumference is an important factor that cannot be ignored. Weight or BMI is not sufficient for many reasons described in details in literature. Hence, the authors should include Waist of both groups and baseline and at end of follow-up.
Response: We really apologize from the reviewer and keenly confident in her/his understanding, but the waist circumference values are not available. However we added this clearly as a limitation of the study (Page 6, paragraph 4).
2) The baseline table looks somehow `poor of data´. In other words, the authors are kindly invited to give more additional data in both group at baseline as presence of metabolic syndrome, Diabetes, hypertension, dyslipidaemia, alcohol consume, smoking status, socioeconomic status of both groups. The presence of differences should be then possibly in a multivariate analysis included. Such information is essential to avoid Bias when comparing the both treatment strategies.
Response: Done as suggested, added the presence of type 2 diabetes, hypertension, dyslipidaemia, and sociodemographic status in the Methods (Page 2, paragraph 6 and page 3, paragraph 2) and Results (Page 3, paragraph 6) sections as well as in the Table 1 (Page 4). We found no differences between these variables among groups.
3) Since the analysis was focused on weight loss as absolute value then it is expected to have analysis of body composition (measured by BIA or DXA). Therefore what was the value of fat mass or fat free mass in these two categories at the start point and end of follow up. This point is also essential to avoid false conclusions about weight loss.
Response: No body composition measurements are available. We added this in the limitation section (Page 6, paragraph 4).
Round 2
Reviewer 2 Report
The authors had addressed the suggested comments.The paper is improved.